# Antiviral kinetics of tenofovir alafenamide and tenofovir disoproxil fumarate over 24 weeks in women of childbearing potential with chronic HBV

Calvin Q. Pan[1,2]*, Ting-Tsung Chang[3], Si Hyun Bae[4], Maurizia Brunetto[5], Wai-Kay Seto[6], Carla S. Coffin[7], Susanna K. Tan[8], Shuyuan Mo[8], John F. Flaherty[8], Anuj Gaggar[8], Mindie H. Nguyen[9], Mustafa Kemal Çelen[10], Alexander Thompson[11], Edward J. Gane[12]

1 Center for Liver Diseases, Beijing Ditan Hospital, Capital Medical University, Beijing, China, 2 NYU Langone Medical Center, New York, New York, United States of America, 3 National Cheng Kung University Hospital, Tainan, Taiwan, 4 College of Medicine, The Catholic University of Korea, Seoul, Republic of Korea, 5 University of Pisa, Pisa, Italy, 6 Department of Medicine, The University of Hong Kong, Hong Kong, Hong Kong, 7 Cumming School of Medicine, University of Calgary, Alberta, Canada, 8 Gilead Sciences, Inc., Foster City, California, United States of America, 9 Stanford University, Palo Alto, California, United States of America, 10 Dicle University School of Medicine, Diyarbakir, Turkey, 11 St. Vincent's Hospital Melbourne and The University of Melbourne, Melbourne, Australia, 12 New Zealand Liver Transplant Unit, Auckland Clinical Studies, Auckland, New Zealand

* Panc01@NYU.edu

**Data Availability Statement:** Gilead shares anonymized Individual Patient Data (IPD) upon request or as required by law and/or regulation

## Abstract

### Background/Purpose

Use of tenofovir disoproxil fumarate (TDF) improves patient outcomes in preventing mother-to-child transmission (pMTCT) of the hepatitis B virus (HBV) in mothers with chronic HBV and high viral loads. Given the lack of data for tenofovir alafenamide (TAF) in pMTCT, rates of early viral suppression with TAF and TDF were evaluated in women of childbearing potential (WOCBP) participating in 2 randomized, double-blind, Phase 3 studies in chronic HBV.

### Methods

In a patient subset meeting WOCBP criteria and with baseline HBV DNA >200,000 IU/mL, rates of viral suppression with TAF or TDF in achieving the target of HBV DNA <200,000 IU/mL at weeks 12 and 24 were assessed. Multivariate logistic regression was used to identify factors predictive of failure to suppress HBV DNA to the target level.

### Results

In 275 of 1298 (21%) patients meeting WOCBP criteria with high viral load, 93% and 96% had HBV DNA <200,000 IU/mL at weeks 12 and 24, respectively. Results for TAF (n = 194) vs TDF (n = 81) treatment were similar at weeks 12 and 24 (94% vs. 90% and 97% vs. 93%), respectively. High baseline HBV DNA level, genotype D infection, and prior interferon

with qualified external researchers. Approval of such requests is at Gilead's discretion and is dependent on the nature of the request, the merit of the research proposed, the availability of the data, and the intended use of the data. Data requests should be sent to datarequest@gilead. com. The authors of this study did not receive any special privileges in accessing the data from Gilead that other researchers would not receive.

**Funding:** Funding for this study was provided by Gilead Sciences, Inc. No specific grant associated with this study, but CQP, T-TC, SHB, MB, W-KS, CSC, MHN, MKC, AT, and EJG served as investigators of the clinical trial from which this secondary analysis is derived. The funder, Gilead Sciences, participated in study design, data collection and analysis, decision to publish, and preparation of the manuscript.

**Competing interests:** Calvin Pan has received research grants from Gilead Sciences (Gilead) and Merck. He also serves as a consultant or advisor for Gilead, and speakers' bureau for Gilead, Abbvie, and Intercept. Ting-Tsung Chang and Si Hyun Bae declare no conflicts of interest. Wai Kay Seto serves on the advisory boards of Gilead, Abbvie and CSL Behring, and is a speaker for Gilead, AbbVie and Mylan. Maurizia Brunetto has received research grants from AbbVie, BMS, MSD, has served on advisory Boards for AbbVie, Gilead, Janssen; Roche, and has served as a speaker for AbbVie, Gilead, MSD. Carla S. Coffin has served as an investigator or received research grants from GlaxoSmithKline, Gilead, Arbutus Biopharma, Bristol-Myers Squibb, has received educational grants from Merck, Gilead, Janssen, has served on advisory boards for Merck, Gilead, GlaxoSmithKline, has served on Clinical Trial and Publication Committee for Spring Bank Pharmaceuticals, and has participated as a primary investigator in clinical trials for Gilead, Spring Bank, Transgene, and Janssen. The following authors are employees of Gilead Sciences and hold stock interest in the company: Susanna Tan, Shuyuan Mo, John Flaherty, and Anuj Gaggar. Mindie H. Nguyen has received research support from Gilead and Pfizer, has served on advisory boards and/or as a consultant for Novartis, Spring Bank, Janssen, Gilead, Eisai, Bayer. Exact Science, and LAM. Mustafa Kemal Çelen has no conflicts of interest to disclose. Alexander Thompson has received research support from Gilead, AbbVie, and Merck, and has served on advisory boards and/or as a consultant for Gilead, AbbVie, Merck, BMS, Eisai. Edward J. Gane has served as a consultant or advisor for Gilead, AbbVie, Roche, Janssen and at speakers' bureaus for Gilead and AbbVie. This does

(week 24 only) were predictive of failure to achieve the target level. Both treatments were well tolerated with TAF showing less impact on renal and bone parameters.

## Conclusions

In WOCBP with high VL, no differences were found between TAF and TDF in reducing HBV DNA to levels associated with lower transmission risk. These data support ongoing studies of TAF for pMTCT.

## Introduction

Worldwide, up to 2 billion people worldwide have become infected with the hepatitis B virus (HBV), with 257 million individuals estimated to have chronic infection (i.e. hepatitis B surface antigen [HBsAg] seropositivity for >6 months) [1, 2]. Although a safe and effective vaccine has been available for many years, vertical transmission of HBV from mother to child still occurs, accounting for approximately 50% of new infections in highly endemic areas and one-third of new cases in areas of lower prevalence [3, 4].

Passive-active infant immunoprophylaxis with hepatitis B immune globulin (HBIG) and the hepatitis B vaccine has greatly reduced the risk of viral transmission from mother to child; however, failures still happen with this approach, particularly when maternal viral load is high at time of delivery [4–7]. In fact, a serum HBV DNA level of 200,000 IU/mL ($2 \times 10^5$ IU/mL) or greater at delivery is identified as a major risk factor for vertical transmission of HBV [5–7]. Notably, in resource-poor settings where access to HBV DNA assays is unavailable, determination of HBeAg seropositivity can be used as an alternative means to determine eligibility for antiviral prophylaxis [8].

Several antiviral agents, notably lamivudine [9], telbivudine [9, 10], and tenofovir disoproxil fumarate [9, 11–13] have been shown to reduce rates of mother-to-child transmission (MTCT) in pregnant women with chronic HBV infection and high viral loads when given in addition to standard immunoprophylaxis. These findings have been confirmed in a recent systematic review and meta-analysis [14]. Of these, TDF is preferred for prevention of MTCT (pMTCT) in all treatment guidelines given its high potency, proven efficacy, and acceptable safety profile in controlled trials [4, 15–18]. For women with chronic active hepatitis B requiring treatment who are, or desire to become pregnant, TDF is the preferred choice by all major guidelines. For women with chronic HBV not meeting treatment criteria, recommendations for when to initiate TDF during pregnancy for prevention of HBV transmission, and what constitutes a high viral load, differs somewhat among guidelines. In pregnant women with HBV DNA >200,000 IU/mL, the American Association for the Study of Liver Diseases (AASLD) recommends TDF during the third trimester (weeks 28–32) with discontinuation at time of delivery or up to 4 weeks postpartum [15]. The European Association for the Study of the Liver (EASL) recommends TDF when HBV DNA is >200,000 IU/ml (or HBsAg >4 $\log_{10}$ IU/mL) starting at weeks 24–28 weeks of gestation and continuing for up to 12 weeks post-delivery [16]. The Asian Pacific Association for the Study of the Liver (APASL) recommends initiation of TDF (or telbivudine as an alternative) at 28–32 weeks of gestation if the mother's HBV DNA level is >6-7 $\log_{10}$ IU/mL with continued treatment until birth [17], while the China algorithm for interrupting MTCT recommends TDF (or telbivudine) to be started at gestational week 24–28 in women with HBV DNA >2 x $10^6$ $\log_{10}$ IU/mL and discontinuation of therapy at delivery [4]. The World Health Organization (WHO) recommends that pregnant

not alter our adherence to PLOS ONE policies on sharing data and materials.

**Abbreviations:** AASLD, American Association for the Study of Liver Diseases; ALT, alanine aminotransferase; APASL, Asian Pacific Association for the Study of the Liver; BMI, body mass index; CrCl, creatinine clearance; eGFR, estimated glomerular filtration rate; HBV, hepatitis B virus; TAF, tenofovir alafenamide; TDF, tenofovir disoproxil fumarate; ULN, upper limit of normal.

women testing positive for HBV infection (HBsAg-positive) with an HBV DNA $\geq 5.3 \log_{10}$ IU/mL ($\geq 200,000$ IU/mL) receive TDF prophylaxis from the 28th week of pregnancy until at least birth for pMTCT [18]. Taken collectively, currently available recommendations suggest use of TDF when maternal HBV DNA is 200,000 IU/ml ($2 \times 10^5$ IU/mL) or greater, with treatment continuing for 12 to 24 weeks depending on timing of the infant birth and whether treatment is continued post-partum.

Tenofovir alafenamide, a novel prodrug of tenofovir, has demonstrated antiviral efficacy non-inferior to that of TDF in Phase 3 studies in chronic HBV patients with elevated HBV DNA and alanine aminotransferase (ALT) levels [19–21]. Because of its greater plasma stability, TAF is given in a lower dose than TDF (25 mg vs 300 mg) yet achieves high intracellular levels of the active form of tenofovir within hepatocytes, while plasma concentrations of tenofovir are approximately 90% lower than with TDF [22, 23]. The unique pharmacologic profile of TAF supports improved renal and bone safety when compared to TDF in comparative trials [19–21]. TAF was approved for treatment of chronic HBV in November 2016, by the US Food and Drug Administration, and is currently approved in over 70 countries worldwide. Although clinical data are limited, TAF has not shown adverse embryo-fetal effects in animals and would be predicted to have a safety profile comparable to that of TDF when used during pregnancy, given that these two prodrugs share the same active form [24]. During the clinical development of TAF, pregnant women were excluded from participation in clinical trials, and if pregnancy occurred, study treatment was discontinued. Several trials evaluating TAF for pMTCT of HBV are underway with results expected within a few years. Thus, to better understand the role for TAF in pMTCT, we performed an analysis of the kinetics of viral load reduction in women with chronic HBV and high baseline viral loads (>200,000 IU/mL) who were of child bearing potential (WOCBP) and treated with TAF or TDF in 2 Phase 3 studies [19, 20].

## Methods

### Study design and participants

This was a secondary analysis of patients enrolled in 2 prospective, randomized, double-blind, Phase 3 studies comparing TAF vs TDF in chronic HBV [19–21]. The original Phase 3 clinical trials were approved by all participating institutional review boards and ethics committees. One study (Study GS-US-320-0110; NCT0190471) enrolled a total of 873 HBeAg-positive patients [19], while the other (Study GS-US-320-0108; NCT01940341) included 425 HBeAg-negative patients, otherwise the studies were identical in design [20]. Detailed eligibility and exclusion criteria and design of these studies are available in the respective publications. In brief, males and nonpregnant females, 18 years of age or older with screening HBV DNA >20,000 IU/mL and ALT >60 U/L (males) or >38 (females) U/L, without cirrhosis or with compensated cirrhosis, and with estimated creatinine clearance (eGFR) >50 mL/min were eligible. Patients coinfected with hepatitis C, hepatitis D, or HIV, those with clinical or laboratory evidence of decompensated liver disease, or evidence of hepatocellular carcinoma were excluded.

WOCBP included females 18 to 49 years of age based on the World Health Organization definition of reproductive age [25] who were identified by investigators as being of childbearing potential, and without history of hysterectomy, bilateral oophorectomy, or ovarian failure. For assessment of viral kinetics during treatment, only WOCBP with HBV DNA levels $\geq 200,000$ IU/mL ($2 \times 10^5$ IU/mL) at baseline were included.

### Endpoints

The primary effectiveness endpoints for this analysis were the proportions with HBV DNA <200,000 IU/mL at week 12 and week 24 by treatment assignment (TAF or TDF). Several

secondary viral endpoints were explored including the proportions with viral decline to <20,000 and <29 IU/mL at weeks 12 and 24, the subsets within each group with baseline HBV DNA ≥ or <8 $\log_{10}$ IU/mL who achieved HBV DNA <200,000 IU/mL at weeks 12 and 24, and responses in HBeAg-positive vs HBeAg-negative WOCBP. Additionally, ALT normalization was assessed in patients with baseline ALT levels above upper limit of normal (ULN). Safety assessments included in the original study protocols were evaluated in WOCBP by reported adverse events, serious adverse events, and lab abnormalities through week 24. Other safety parameters were change in eGFR by the Cockcroft-Gault method, and percent change in bone mineral density (BMD) at week 24 by dual energy X-ray absorptiometry (DXA) scans performed at hip and lumbar spine.

## Assessments

Blood samples were collected every 4 weeks and plasma HBV DNA was determined by COBAS Taqman HBV Test for Use with the High Pure System (Roche Molecular Diagnostics, Pleasanton, California, USA) having a lower limit of quantitation of 29 IU/mL. ALT normalization was assessed using ULN cutoffs for females by 2016 AASLD (≤19 U/L) and central laboratory (≤34 U/L) criteria.

## Statistical analyses

Included in the statistical analyses were females that met WOCBP criteria with baseline HBV DNA levels ≥200,000 IU/mL who were randomized and had received at least 1 dose of study medication. Viral suppression and ALT normalization analyses were performed by modified intention-to-treat, missing equals failure. Differences in proportions with 95% confidence intervals (CI) were determined for the rates of viral suppression at the weeks 12 and 24 endpoints (HBV DNA <200,000, <20,000, and <29 IU/mL) by Mantel-Haenszel Tests adjusted by baseline viral load (≥ and < 8 $\log_{10}$ IU/mL) and oral antiviral treatment status (naïve and treatment experienced), and p values were calculated from Cochran- Mantel-Haenszel Tests stratified by baseline viral load and antiviral treatment status. Evaluation of risk factors for failure to achieve the viral suppression target (<200,000 IU/mL at weeks 12 and 24) were determined by univariate (UV) and multivariate (MV) analyses. Variables included in the UV analysis are provided in S1 Table. Factors with a p-value <0.15 by UV analysis were included in the MV stepwise model. P-values for change in eGFR from baseline were determined from the 2-sided Wilcoxon ranked-sum test, and when comparing mean percent change in BMD, an ANOVA model was used which included treatment as a fixed effect. Only patients with non-missing data were included in eGFR and BMD treatment comparisons.

## Results

### Patient population

Of the total enrollment of 1298 patients across the 2 Phase 3 studies, 479 (37%) were female, and 275 (21%) patients met WOCBP criteria with HBV DNA >200,000 IU/mL. Of the 275 patients, 194 (71%) and 81 (29%) received TAF and TDF, respectively. At baseline the two groups of were generally well matched (Table 1), having a median age of 35 years, 81% Asian, and the majority (52%) were genotype C. Over 80% were HBeAg-positive, with a slightly higher proportion in the TDF vs TAF group. Mean HBV DNA levels were similar at baseline; 40% overall had a level ≥8 $\log_{10}$ IU/mL. A small percentage (<8%) had cirrhosis by history or Fibrotest score ≥0.75; nearly all had ALT >ULN by both criteria. Only a small proportion (<5%) had comorbidities of hypertension, cardiovascular disease, and/or diabetes mellitus.

**Table 1. Patient demographics and baseline characteristics in the subset of WOCBP (studies GS-US-320-0110 and GS-US-320-0108; week 96 integrated safety analysis set; reference 21).**

| | TAF (n = 194) | TDF (n = 81) | Total (N = 275) |
|---|---|---|---|
| Median (range) age, yr. | 35 (18–49) | 36 (18–48) | 35 (18–49) |
| Race, n (%) | | | |
| Asian | 158 (81) | 65 (80) | 223 (81) |
| Black or African American | 1 (<1) | 0 | 1 (<1) |
| White | 35 (18) | 16 (20) | 51 (19) |
| Mean (SD) BMI, kg/m$^2$ | 23.0 (4.45) | 22.5 (3.96) | 22.9 (4.31) |
| BMI ≥30 kg/m$^2$, n (%) | 12 (6) | 3 (4) | 15 (5) |
| HBeAg positive, n (%) | 156 (80) | 71 (88) | 227 (83) |
| Mean (SD) HBsAg level, log$_{10}$ IU/mL | 3.99 (0.66) | 4.03 (0.63) | 4.01 (0.65) |
| HBsAg >4 log$_{10}$ IU/mL, n (%) | 99 (51) | 40 (49) | 139 (51) |
| Mean (SD) HBV DNA, log$_{10}$ IU/mL | 7.6 (1.05) | 7.6 (1.04) | 7.6 (1.05) |
| Baseline HBV DNA, n (%) | | | |
| <7 log$_{10}$ IU/mL | 51 (26) | 23 (28) | 74 (27) |
| ≥7 to < 8 log$_{10}$ IU/mL | 69 (36) | 23 (28) | 92 (33) |
| ≥8 log$_{10}$ IU/mL | 74 (38) | 35 (43) | 109 (40) |
| Median (Q1, Q3) ALT, U/L | 77 (50, 111) | 67 (53, 116) | 74 (51, 113) |
| ALT, central laboratory | | | |
| ≤ULN | 17 (9) | 3 (4) | 20 (7) |
| >ULN to 5 x ULN | 151 (78) | 65 (80) | 216 (79) |
| >5 x ULN to 10 x ULN | 21 (11) | 9 (11) | 30 (11) |
| >10 x ULN | 5 (3) | 4 (5) | 9 (3) |
| ALT, AASLD[a] | | | |
| ≤ULN | 1 (<1) | 0 | 1 (<1) |
| >ULN to 5 x ULN | 131 (68) | 51 (63) | 182 (66) |
| >5 x ULN to 10 x ULN | 42 (22) | 20 (25) | 62 (23) |
| >10 x ULN | 20 (10) | 10 (12) | 30 (11) |
| HBV genotype, n (%) | | | |
| A | 10 (5) | 2 (2) | 12 (4) |
| B | 42 (22) | 16 (20) | 58 (21) |
| C | 99 (51) | 44 (54) | 143 (52) |
| D | 42 (22) | 19 (23) | 61 (22) |
| E | 1 (<1) | 0 | 1 (<1) |
| Cirrhosis history, n (%) | | | |
| Yes | 9 (7) | 5 (7) | 14 (7) |
| No | 118 (93) | 64 (93) | 182 (93) |
| Indeterminate/unknown | 67 | 12 | 79 |
| Fibrotest score ≥0.75[b], n (%) | 4/185 (2) | 1/79 (1) | 5/264 (2) |
| Prior treatment experience, n (%) | | | |
| Oral nucleos(t)ide | 34 (18) | 28 (35) | 62 (23) |
| Interferons | 23 (12) | 11 (14) | 34 (12) |
| Median (Q1, Q3) eGFR,[c] mL/min | 107 (94, 126) | 103 (89, 127) | 106 (92, 127) |
| Hypertension, n (%) | 7 (4) | 5 (5) | 11 (4) |
| Cardiovascular disease, n (%) | 1 (<1) | 1 (1) | 2 (<1) |

*(Continued)*

**Table 1.** (Continued)

|  | TAF (n = 194) | TDF (n = 81) | Total (N = 275) |
|---|---|---|---|
| Diabetes mellitus, n (%) | 6 (3) | 4 (5) | 10 (4) |

[a]*ULN*, 19 U/L for females and 30 U/L for males.

[b]Suggestive of Metavir fibrosis stage F4 (cirrhosis)

[c]*eGFR*, estimated glomerular filtration rate using the Cockcroft-Gault method.

*ALT*, alanine aminotransferase; *TAF*, tenofovir alafenamide; *TDF*, tenofovir disoproxil fumarate, *ULN*, upper limit of normal range.

## Virologic responses at weeks 12 and 24

Overall, 93% (255/275) and 96% (264/275) of patients had HBV DNA <200,000 IU/mL at weeks 12 and 24, respectively. A slightly higher proportion of TAF- vs TDF-treated patients achieved this endpoint at week 12 (TAF 94% vs TDF 90%; difference in proportion [95% CI]: 2.1% [-5.7% to 9.8%]; p = 0.504) and week 24 (TAF 97% vs TDF 93%; difference in proportion 4.4% [-2.7% to 11.4%]; p = 0.105); but in general, the kinetics of viral suppression were similar between groups over this period (Fig 1). Among patients with high baseline viral load (HBV DNA ≥8 $\log_{10}$ IU/mL), the proportion achieving HBV DNA <200,000 IU/mL was similar between treatments: 88% (65/74) with TAF and 86% (30/35) with TDF at week 12, and 96% (71/74) with TAF and 91% (32/35) with TDF treatment at week 24 (Fig 2A and 2B).

By week 12, 76% of patients overall (209/275) achieved HBV DNA < 20,000 IU/mL with similar response rates between treatments (TAF 76% vs TDF 75%; difference in proportion -1.9% [-12.9% to 9.1%]; p = 0.727); by week 24, 91% (251/275) achieved this endpoint with a higher response for TAF vs TDF (93% vs 86%; difference in proportion 4.7% [-4.0% to 13.5%]; p = 0.215) (Fig 1B). Full suppression rates (HBV DNA <29 IU/mL) were lowest overall and similar between groups: TAF 9% (18/194), TDF 9% (7/81) at week 12 (difference in proportion -0.6% [-8.7% to 7.5%]; p = 0.872), and TAF 42% (81/194), TDF 47% (38/81) at week 24 (difference in proportion -8.5% [-20.3% to 3.3%]; p = 0.162) (Fig 1C).

Among HBeAg-positive patients, the target of HBV DNA <200,000 IU/mL was achieved in 92% (144/156) and 90% (64/71) of TAF and TDF patients at week 12, respectively; at week 24, 97% (151/156) and 93% (66/71) of TAF and TDF patients, respectively, met this endpoint (Fig 2C). In the smaller subset of HBeAg-negative patients (n = 48), nearly all TAF and TDF patients achieved the target level (98% at weeks 12 and 24) (Fig 2D).

UV analysis identified several factors meeting criteria for inclusion in the MV model (S1 Table); however, higher baseline HBV DNA level and genotype D infection were the only predictors for early failure in achieving the target level (Table 2). Among patients with HBV DNA

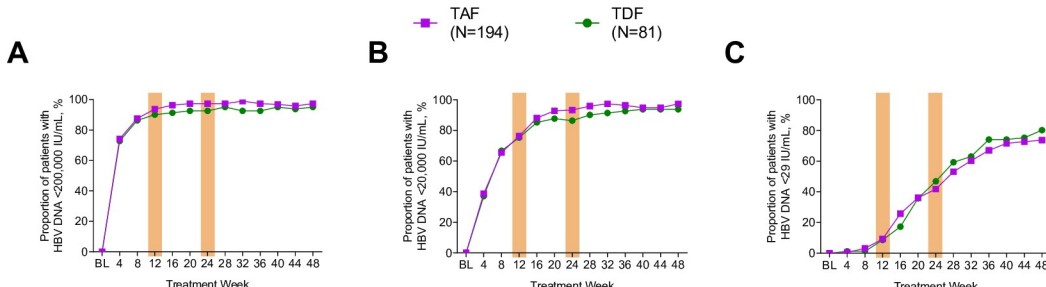

**Fig 1. Rates of viral decline to <200,000 IU/mL, <20,000 IU/mL, and <29 IU/mL.** TAF, tenofovir alafenamide; TDF, tenofovir disoproxil fumarate.

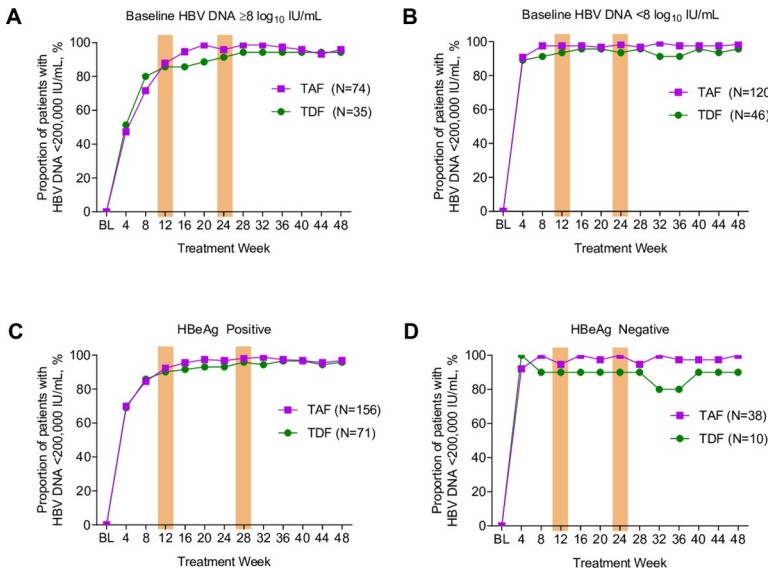

**Fig 2. Rates of viral decline to <200,000 IU/mL by select baseline characteristics.** Baseline Viral Load (A) $\geq 8$ $\log_{10}$ IU/mL and (B) $<8$ $\log_{10}$ IU/mL, (C) HBeAg-positive and (D) HBeAg-negative patients. TAF, tenofovir alafenamide; TDF, tenofovir disoproxil fumarate.

$\geq 8$ $\log_{10}$ IU/mL at baseline, the week 12 response rate overall was 73% for those with genotype D infection compared to 94% for non-genotype D patients. At week 24, only prior interferon treatment was predictive of target failure (**Table 2**).

## ALT normalization

Overall, ALT normalization was observed in 44% of patients at 12 weeks (TAF 45%, TDF 40%), and in 64% of patients at 24 weeks (TAF 65%, TDF 63%) by central lab criteria (**Table 3**). Rates of ALT normalization were lower and similar between treatments when assessed by the AASLD criteria (**Table 3**).

## Safety

In the WOCBP population, similar proportions experienced at least 1 adverse event (TAF 71%, TDF 68%) over 24 weeks (**Table 4**). Most adverse events were mild or moderate in severity with no patients discontinuing study treatment for an advent event. The most common adverse events ($\geq 6\%$) were nasopharyngitis, headache, upper respiratory tract infection, cough, upper abdominal pain, and dyspepsia.

**Table 2. Multivariate analysis: Predictors of HBV DNA $\geq$200,000 IU/mL.**

|  | Week | |
| --- | --- | --- |
|  | **12** | **24** |
| Baseline HBV DNA ($\log_{10}$ IU/mL) | 5.8 (1.9, 17.6) p = 0.0021 | - |
| HBV genotype D | 4.3 (1.3,14.7) p = 0.0203 | - |
| Prior interferon treatment | - | 14.5 (1.3, 164.5) p = 0.0309 |

Values are reported as odds ratio (95% CI).

*HBV*, hepatitis B virus.

**Table 3. ALT normalization at weeks 12 and 24 in the subset of WOCBP (studies GS-US-320-0110 and GS-US-320-0108, week 96 full analysis set; reference 21).**

|  | TAF | TDF | Total |
|---|---|---|---|
| Central lab[a], n/n (%) |  |  |  |
| Week 12 | 80/177 (45) | 31/78 (40) | 111/255 (44) |
| Week 24 | 115/177 (65) | 49/78 (63) | 164/255 (64) |
| AASLD,[b] n/n (%) |  |  |  |
| Week 12 | 16/193 (8) | 7/81 (9) | 23/274 (8) |
| Week 24 | 40/193 (21) | 18/81 (22) | 58/274 (21) |

*AASLD*, American Association for the Study of Liver Diseases; *ALT*, alanine aminotransferase; *TAF*, tenofovir alafenamide; *TDF*, tenofovir disoproxil fumarate.
[a]Upper limit of normal (*ULN)*, 34 U/L for females.
[b]*ULN*, *19* U/L for females.

Four patients in the TAF group (2%) experienced 1 serious adverse event each of anemia, appendicitis, pneumonia, and pyrexia; none of these events was treatment-related or required treatment interruption or discontinuation. The most commonly reported Grade 3 laboratory abnormalities included blood in the urine and elevations in serum ALT (**Table 4**).

**Table 4. Adverse events through 24 weeks in the subset of WOCBP (studies GS-US-320-0110 and GS-US-320-0108; week 96 integrated safety analysis set; reference 21).**

|  | TAF (n = 194) | TDF (n = 81) |
|---|---|---|
| Any adverse event | 138 (71) | 55 (68) |
| Any study drug–related adverse event | 38 (20) | 17 (21) |
| Any Grade 3 or 4 adverse event[a] | 3 (2) | 1 (1) |
| Any serious adverse event[a] | 4 (2) | 0 |
| Any serious adverse event related to study treatment | 0 | 0 |
| Any adverse event leading to discontinuation of study treatment | 0 | 0 |
| Deaths, n | 0 | 0 |
| Common AEs (≥6% in either group) |  |  |
| Nasopharyngitis | 24 (12) | 4 (5) |
| Headache | 16 (8) | 7 (9) |
| Upper respiratory tract infection | 13 (7) | 4 (5) |
| Cough | 12 (6) | 3 (4) |
| Nausea | 12 (6) | 6 (7) |
| Abdominal pain (upper) | 11 (6) | 2 (2) |
| Dyspepsia | 8 (4) | 5 (6) |
| Lab Abnormalities (≥2% in either group) |  |  |
| Occult Blood, Grade 3 | 35 (18) | 16 (20) |
| Urine Erythrocytes, Grade 3 | 28 (14) | 19 (23) |
| Alanine Aminotransferase, >5 × ULN | 18 (9) | 7 (9) |
| Aspartate Aminotransferase, >5 × ULN | 5 (3) | 6 (7) |
| Creatine Kinase, ≥10 × ULN | 5 (3) | 0 |
| Urine Glucose, Grade 3 | 3 (2) | 0 |
| Hemoglobin, 7 to <9 g/dL | 3 (2) | 2 (3) |

Data are presented as n patients (%). *AE*, adverse event; *TAF*, tenofovir alafenamide; *TDF*, tenofovir disoproxil fumarate.
[a]No reported Grade 3 or 4, or serious AEs were considered to be related to study drug.

Median eGFR remained stable over the treatment period in the TAF group, whereas declines were observed in patients receiving TAF (S1 Fig). In TDF-treated patients, median (Q1, Q3) eGFR declined at week 12 compared with an increase on TAF treatment (-6.0 [-13.2, 2.4] vs +1.8 [-7.6, 8.7] mL/min; p<0.001); however, at week 24, eGFR change was similar between groups. Mean (SD) percent changes in BMD from baseline showed greater decreases with TDF vs. TAF at spine (-2.33% [2.49] vs -0.38% [2.30]; p<0.001) and hip (-0.77% [1.98] vs -0.17% [1.75]; p = 0.017) at week 24.

## Discussion

Antiviral therapy with TDF given during the third trimester of pregnancy has been shown to reduce the risk of HBV transmission from mother to infant in several controlled clinical trials [9, 11–13], and in a recent systematic review and meta-analysis [14]. Data regarding the efficacy and safety of TAF for pMTCT of HBV are currently lacking. In this secondary analysis of early viral kinetics in a well characterized cohort of WOCBP with chronic HBV and high baseline viral load, we found that treatment with TAF was similarly effective to TDF in reducing plasma HBV DNA to a threshold associated with minimal or low risk of mother-to-child transmission of the virus to infants who are also given passive-active immunoprophylaxis [12]. After 12 weeks of treatment with TAF or TDF, 90% or more had a reduction in plasma HBV DNA to <200,000 IU/mL, and by 24 weeks, an even higher proportion of patients achieved this goal. These results also suggest that at least 12 weeks treatment is optimal to ensure most pregnant women with high viral loads achieve the target HBV DNA level by the approximate time of delivery. This is of relevance given that certain guidelines for pMTCT of HBV recommend starting TDF later than week 28, and while 40 weeks is typical, the timing of delivery does vary with up to 12.5% of births occurring before gestation week 37 [26]. Our data are supportive of guidelines by EASL and China which recommend initiation of treatment earlier at gestational weeks 24–28 [4, 16]. Waiting to initiate treatment between gestational weeks 28–32, as per the APASL, AASLD, and recently issued WHO guidelines [15, 17, 18], might be suboptimal for certain women, particularly in the setting of very high HBV DNA levels and/or preterm delivery.

Several studies support the level of maternal viremia being a critical factor when considering the timing of antiviral therapy pMTCT of HBV. By multivariate analysis, we identified 2 factors—higher baseline HBV DNA level and infection with genotype D as predictors for failure to achieve the viral suppression target at 12 weeks. High baseline viral load is widely known to affect rates of HBV DNA suppression as previously reported for TDF and ETV in patients with chronic HBV [27, 28]. Although less is known about the impact of HBV genotype on kinetics of viral suppression, we have previously reported infection with genotype D is associated with a slower rate of viral decline with TAF and TDF treatment, although the reason for this finding is unclear [29]. There are many areas globally where genotype D infection is common and wherein perinatal transmission remains an important source of HBV acquisition (e.g. parts of Africa, the Indian subcontinent, and Southern Europe [30]).

The viral kinetic data from this analysis correlate well with published studies that evaluated the association between maternal HBV DNA decline and infant outcomes when TDF is used for pMTCT of HBV [11–13]. In a randomized, controlled study in HBeAg-positive pregnant women with high viral load, TDF treatment started at pregnancy week 30-32 significantly reduced the rate of viral transmission to infants compared with controls (i.e. same infant immunoprophylaxis regimen without antiviral use in mothers): 5% vs 18%, respectively; p = 0.007 [12]. Substantially more mothers given TDF achieved the suppression target of HBV DNA <200,000 IU/mL at delivery vs controls (68% vs 2%; p<0.001), while in TDF-treated

mothers the proportion with HBV DNA > 200,000 IU/mL at delivery was 1.5-fold greater when baseline HBV DNA was higher (> vs ≤8 $\log_{10}$ IU/mL; 39% vs 25%; p = 0.19). In a randomized, controlled study which compared TDF to placebo for pMTCT of HBV (passive-active immunoprophylaxis regimen given to all infants), pregnant women randomized to TDF (median gestational age 28 weeks) showed no HBV transmissions compared to 2% (3 infants) with placebo (p = 0.12) [13]. At time of delivery, 12% of TDF-treated mothers had HBV DNA >200,000 IU/mL as compared with 90% in the placebo group. In a controlled study of TDF vs no TDF treatment along with infant passive-active immunoprophylaxis [11], of 62 mothers who received TDF (mean screening viral load 8.25 $\log_{10}$ IU/mL) starting at 30–32 weeks gestation, 61 (98%) had HBV DNA <6 $\log_{10}$ IU/mL at the time of delivery, with a good correlation (correlation coefficient 0.8098; p<0.0001) between duration of TDF treatment and $\log_{10}$ reduction in HBV DNA levels, while TDF treatment was associated with a lower risk of HBsAg seropositivity in infants at 6 months (1.54% vs 10.71%; p = 0.048).

In the present analysis, treatment with TAF or TDF was safe and well tolerated and no patients discontinued treatment for an adverse event. WOCBP who received TAF showed only minimal declines in estimated glomerular filtration rate compared to greater declines with TDF, particularly at 12 weeks. Those receiving TAF also showed smaller mean percent declines in hip and spine BMD at 24 weeks. Whether the better bone and renal safety profile we observed for TAF vs TDF in WOCBP is confirmed in clinical studies with TAF for pMTCT of HBV remains to be determined.

There are several limitations to the present analysis. This was a descriptive, secondary analysis of viral suppression rates and safety in a subset of WOCBP using data previously collected from 2 large Phase 3 studies for CHB [25]. In applying the WHO definition, it is possible some postmenopausal women may have been included, and conversely, some WOCBP not fitting the WHO definition, may have been excluded. Also, the median age of the women included in our analysis (35 years) may be higher than the age of most pregnant women in many countries. Further, the study population was drawn from studies of patients with chronic active HBV who met criteria to initiate oral antiviral treatment by all major guidelines. This study population differs from the population for antenatal antiviral prophylaxis to prevent mother-to-child transmission–untreated women who are HBeAg-positive in the immunotolerant or "chronic HBV infection" phase of chronic HBV [16]. Thus, given these differences, it is unclear whether the rates of viral suppression we observed here would directly apply to the population typically given antivirals for pMTCT of HBV. Lastly, the physiological state of pregnancy can alter pharmacokinetics of some drugs, including antivirals [30–32]. Studies in pregnant women monoinfected with HBV [30] or HIV [31] receiving TDF have reported approximately 20% lower tenofovir exposures, particularly during the third trimester. The impact of these small changes in tenofovir pharmacokinetics on therapeutic response in pMTCT is not expected to be important, while the impact of pregnancy on TAF pharmacokinetics is presently unknown.

In conclusion, in WOCBP with high viral load, treatment with TAF or TDF for 12 and 24 weeks showed similar effectiveness in reducing HBV DNA to levels associated with minimal viral transmission to infants who receive complete HBV passive-active immunoprophylaxis. Findings from this analysis will inform present and future studies evaluating TAF for the prevention of mother-to-child transmission of the hepatitis B virus.

## Supporting information

**S1 Table. Univariate analysis.** Predictors of HBV DNA ≥ 200,000 IU/mL at Week 12.
(DOCX)

**S1 Fig. Changes from baseline in estimated Glomerular Filtration Rate (eGFR) by the Cockcroft-Gault method.** SD, standard deviation; TAF, tenofovir alafenamide; TDF, tenofovir disoproxil fumarate. $^*$p$<$0.05, †p$<$0.001.
(DOCX)

# Acknowledgments

We thank the patients and their families as well as the study-site personnel. Assistance with additional statistical analysis was provided by Hongyuan Wang, PhD, of Gilead Sciences. Writing assistance was provided by Jennifer King, PhD, of August Editorial. Editorial assistance was provided by Sandra Chen, of Gilead Sciences. All authors approved the final version of the article, including the authorship list.

# Author Contributions

**Conceptualization:** Calvin Q. Pan, John F. Flaherty, Anuj Gaggar.

**Data curation:** Calvin Q. Pan.

**Formal analysis:** Shuyuan Mo.

**Investigation:** Calvin Q. Pan, Ting-Tsung Chang, Si Hyun Bae, Maurizia Brunetto, Wai-Kay Seto, Carla S. Coffin, Susanna K. Tan, Mindie H. Nguyen, Mustafa Kemal Çelen, Alexander Thompson, Edward J. Gane.

**Methodology:** Calvin Q. Pan.

**Supervision:** Anuj Gaggar.

**Validation:** Calvin Q. Pan.

**Visualization:** Calvin Q. Pan.

**Writing – original draft:** John F. Flaherty.

**Writing – review & editing:** Calvin Q. Pan, Ting-Tsung Chang, Si Hyun Bae, Maurizia Brunetto, Wai-Kay Seto, Carla S. Coffin, Susanna K. Tan, Shuyuan Mo, John F. Flaherty, Anuj Gaggar, Mindie H. Nguyen, Mustafa Kemal Çelen, Alexander Thompson, Edward J. Gane.

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
