## [Decision Letter · Decision Letter 0]

27 Jan 2021

PONE-D-20-22518

Antiviral Kinetics of Tenofovir Alafenamide and Tenofovir Disoproxil Fumarate over 24 Weeks in Women of Childbearing Potential with Chronic HBV

PLOS ONE

Dear Dr. Pan,

Thank you for submitting your manuscript to PLOS ONE. After careful consideration, we feel that it has merit but does not fully meet PLOS ONE’s publication criteria as it currently stands. Therefore, we invite you to submit a revised version of the manuscript that addresses the points raised during the review process.

Your manuscript has been evaluated by two external experts who have highlighted concerns regarding the statistical analysis, contextualisation within the existing literature and other aspects of the methodology. Please ensure you address all of these aspects as part of your revisions.

We look forward to receiving your revised manuscript.

Kind regards,

Joseph Donlan

Senior (Staff) Editor

PLOS ONE

Journal Requirements:

2.) Thank you for stating the following in the Competing Interests section:

'Calvin Pan has received research grants from Gilead and Merck. He also serves as a consultant or advisor for Gilead, and speakers’ bureau for Gilead, Abbvie, and Intercept. Ting-Tsung Chang and Si Hyun Bae declare no conflicts of interest. Wai Kay Seto serves on the advisory boards of Gilead, Abbvie and CSL Behring, and is a speaker for Gilead, AbbVie and Mylan. Maurizia Brunetto has received research grants from AbbVie, BMS, MSD, has served on advisory Boards for AbbVie, Gilead, Janssen; Roche, and has served as a speaker for AbbVie, Gilead, MSD. Carla S. Coffin has served as an investigator or received research grants from GlaxoSmithKline, Gilead, Arbutus Biopharma, Bristol-Myers Squibb, has received educational grants from Merck, Gilead, Janssen, has served on advisory boards for Merck, Gilead, GlaxoSmithKline, has served on CTPC committees for Springbank, and has participated as a primary investigator in clinical trials for Gilead, Springbank, Transgene, and Janssen. The following authors are employees of Gilead Sciences and hold stock interest in the company: Susanna Tan, Shuyuan Mo, John Flaherty, and Anuj Gaggar. Mindie H. Nguyen has received research support from Gilead and Pfizer, has served on advisory boards and/or as a consultant for Novartis, Springbank, Janssen, Gilead, Eisai, Bayer. Exact Science, and LAM. Mustafa Kemal Çelen has no conflicts of interest to disclose. Alexander Thompson has received research support from Gilead, AbbVie, and Merck, and has served on advisory boards and/or as a consultant for Gilead, AbbVie, Merck, BMS, Eisai. Edward J. Gane has served as a consultant or advisor for Gilead, AbbVie, Roche, Janssen and at speakers’ bureaus for Gilead and AbbVie.'

3.) We note that you have indicated that data from this study are available upon request. PLOS only allows data to be available upon request if there are legal or ethical restrictions on sharing data publicly. For information on unacceptable data access restrictions, please see http://journals.plos.org/plosone/s/data-availability#loc-unacceptable-data-access-restrictions.

4.) Please include captions for your Supporting Information files at the end of your manuscript, and update any in-text citations to match accordingly. Please see our Supporting Information guidelines for more information: http://journals.plos.org/plosone/s/supporting-information.

Reviewers' comments:

Reviewer's Responses to Questions

**Comments to the Author**

1. Is the manuscript technically sound, and do the data support the conclusions?

Reviewer #1: Partly

Reviewer #2: Yes

2. Has the statistical analysis been performed appropriately and rigorously? 

Reviewer #1: No

Reviewer #2: Yes

3. Have the authors made all data underlying the findings in their manuscript fully available?

Reviewer #1: No

Reviewer #2: Yes

4. Is the manuscript presented in an intelligible fashion and written in standard English?

Reviewer #1: Yes

Reviewer #2: Yes

5. Review Comments to the Author

Reviewer #1: PONE-D-20-22518: statistical review

SUMMARY. This is a secondary analysis of subjects with chronic hepatitis B, enrolled in two previous studies and respectively treated with tenofovir alafenamide (TAF) and tenofovir disoproxil fumarate (TDF). It compares the proportion of subjects who reach the viral supression target (HBV DNA <200,000 IU/mL) at weeks 12 and 24 of the follow-up period. Supplementary analyses are made by considering alternative viral supression targets in subgroups. In a second part of the study, logistic regression analysis is performed to detect significant risk factors for failure to achieve the viral suppression target. While I have nothing to say about the second part of the study (methods are correct and results are clearly described and appropriately interpreted), I have a couple of concerns about the first part of the analysis: see major issues 1 and 2 below.

MAJOR ISSUES

1) One main finding of the study is that TAF and TDF have a similar effect on viral suppression at weeks 12 and 24, as described at the beginning of page 11. However, it looks like the difference between proportions has not been tested (ot it has, but results have not been reported ...). A two-sample test for proportions can be performed to show that, separately at weeks 12 and 24, proportions are not significately different. Or, perhaps more elegantly, an ANOVA logistic model can be estimated to perform a simultaneous test on both the 12th and the 24th week.

2) Figures 1 and 2 display the cumulative proportion of subjects who reach the suppression target at several weeks. I wonder why we shoud limit our attention to these two weeks when we could compare the two curves as a whole. For example, taking the complementary cumulative proportions (e.g. 1-p_w, where p_w is the cumulative proportion at week w), we obtain two survival curves that can be compared by a nonparametric logrank test, without the need of specific parametric assumptions. The authors should either perform this analysis or, alternatively, explain why weeks 12 and 24 are the only relevant weeks in this study.

Reviewer #2: The manuscript entitled “Antiviral kinetics of TAF and TDF over 24 weeks in women of childbearing potential with chronic HBV” addresses an important topic in optimizing chronic HBV treatment among women of childbearing potential, as pregnant women were previously excluded from participation in clinical trials. The Introduction states that ‘several investigator-sponsored trials evaluating TAF for PMTCT of HBV are underway with results expected within a few years.” It is unclear to this reviewer why it takes several more years for these important trials to complete (given that the follow-up period for HBV is relatively short).

The present study provides reassuring data on early viral suppression in women of childbearing age and, as expected, TAF was well tolerated and showed less impact on renal and bone parameters than TDF. These findings are important and do support the ongoing clinical trials of TAF in pregnant women for HBV PMTCT. The reviewer notes that 4 coauthors are employed by the company (Gilead Sciences) and many coauthors have potential conflicts of interest; however, these potential conflicts have been transparently reported.

Following are several comments and suggestions for the authors:

(1) The authors mention current HBV care and treatment guidelines from the various professional liver societies around the world (with substantial variations noted in these regional guidelines). However, the authors fail to include the WHO HBV guidelines. On World Hepatitis Day (July 2020), WHO published updated guidelines for the PMTCT of HBV – which are highly relevant to the present article.

(2) Two systematic reviews prepared for the latest 2020 WHO HBV guidelines have appeared in Lancet Infectious Diseases (published online August 2020; in hard copy, current issue): Boucheron et al; and Funk et al. These 2 systematic reviews provide relevant summary data for the Introduction and Discussion sections of your paper.

(3) First sentence “Worldwide, over 2 billion are infected with HBV …” (note this doesn’t seem accurately worded; over 2 billion people have become infected at some time during their lifetime). Chronic HBV infections worldwide are estimated by WHO at 257 million persons (see World Hepatitis Report, WHO, Geneva, 2017). The cited paper by Polaris – your reference 2 - is a relative outlier (‘up to 292 million’) among several published HBV modeling papers from 4-5 different groups. Your reference 1 (Schweitzer et al) estimated the number of people living with HBV infection at 240 million. The WHO estimate was a consensus estimate (involving many of the same investigator groups) – this reviewer recommends using the WHO (consensus) estimate.

(4) Intro, page 6, last para: Could you mention anything on the price differential between TAF and TDF? In practice, this will be an important determinant of use.

(5) “Currently approved in over 70 countries worldwide, including China, Japan, S Korea, Taiwan, and Hong Kong.” This seems overly focused on one area of the world. How about Europe, Africa, S America? (or simply delete individual countries).

(6) Page 7, line 3: ‘…same active moiety”. It seems accurately worded in this context. Just keep in mind, many people - like this reviewer - may not be familiar with ‘moiety’.

(7) Page 7, line 5: “Investigator-sponsored trials …” - not sure I understand this. Do you mean “investigator-initiated”? Usually each trial has a financial sponsor (e.g., NIH, a company or a foundation).

(8) Page 8, line: “2 identically designed, prospective, randomized trials…” However, we learn that one trial focused on HBeAg-positive patients and the other on HBeAg-negative patients. Therefore, I think ‘identically designed’ is incorrect.

(9) The study design included women of childbearing potential as ‘females 18-49 years of age’ – with a median age of 35 years. Thus, the average age in your study is higher than the average age of pregnant women in most countries (around 25-26 years of age) though this is changing. Not sure if this 10-year age difference might change any of the findings – I suggest including it among the study limitations.

(10) Discussion: page 13, line 12-13: Does your study provide any data on HBV viral load suppression ‘less than 12 weeks’ after initiation of HBV treatment (either TAF or TDF)? Such data would be very helpful.

(11) Discussion, end of para 1: Again, it would be helpful to include the updated 2020 WHO HBV guidelines in your Discussion. In many countries (e.g., in Africa, Latin America, many countries in Asia outside China), these would be the ones used.

Overall, this is an important paper and the findings among women of childbearing potential appear quite unique in the HBV literature.

Here are a few minor issues:

1) Reference 3 lacks information on year and pages.

2) Table headings need more information: for example, they should include information about the study population/year of the study cohorts included. Currently, the table headings cannot be interpreted or ‘stand on their own’.

6. PLOS authors have the option to publish the peer review history of their article (what does this mean?). If published, this will include your full peer review and any attached files.

Reviewer #1: No

Reviewer #2: No

---

## [Author Response · Author response to Decision Letter 0]

24 Mar 2021

Manuscript ID PONE-D-20-22518: Responses to Editor and Peer Reviewer Comments

EDITORIAL COMMENTS:

Author’s reply

We have reviewed the templates and format the manuscript accordingly. 

2.) Thank you for stating the following in the Competing Interests section:

Author’s reply

The COI statements for all authors have been updated.

3.) We note that you have indicated that data from this study are available upon request. PLOS only allows data to be available upon request if there are legal or ethical restrictions on sharing data publicly. For information on unacceptable data access restrictions, please see http://journals.plos.org/plosone/s/data-availability#loc-unacceptable-data-access-restrictions.

In your revised cover letter, please address the following prompts: a) If there are ethical or legal restrictions on sharing a de-identified data set, please explain them in detail (e.g., data contain potentially identifying or sensitive patient information) and who has imposed them (e.g., an ethics committee). Please also provide contact information for a data access committee, ethics committee, or other institutional body to which data requests may be sent. b) If there are no restrictions, please upload the minimal anonymized data set necessary to replicate your study findings as either Supporting Information files or to a stable, public repository and provide us with the relevant URLs, DOIs, or accession numbers. Please see http://www.bmj.com/content/340/bmj.c181.long for guidelines on how to de-identify and prepare clinical data for publication. For a list of acceptable repositories, please see http://journals.plos.org/plosone/s/data-availability#loc-recommended-repositories.

Author’s reply

As there are legal restrictions against sharing the study data, we have provided the following data availability statement to be included in the manuscript in the cover letter: 

“Gilead shares anonymized Individual Patient Data (IPD) upon request or as required by law and/or regulation with qualified external researchers. Approval of such requests is at Gilead’s discretion and is dependent on the nature of the request, the merit of the research proposed, the availability of the data, and the intended use of the data. Data requests should be sent to datarequest@gilead.com.” 

4.) Please include captions for your Supporting Information files at the end of your manuscript, and update any in-text citations to match accordingly. Please see our Supporting Information guidelines for more information: http://journals.plos.org/plosone/s/supporting-information.

Author’s Reply

We have included captions for supporting information at the end of the manuscript as instructed.

REVIEWER COMMENTS:

Reviewer #1: PONE-D-20-22518: statistical review

SUMMARY. This is a secondary analysis of subjects with chronic hepatitis B, enrolled in two previous studies and respectively treated with tenofovir alafenamide (TAF) and tenofovir disoproxil fumarate (TDF). It compares the proportion of subjects who reach the viral suppression target (HBV DNA <200,000 IU/mL) at weeks 12 and 24 of the follow-up period. Supplementary analyses are made by considering alternative viral suppression targets in subgroups. In a second part of the study, logistic regression analysis is performed to detect significant risk factors for failure to achieve the viral suppression target. While I have nothing to say about the second part of the study (methods are correct and results are clearly described and appropriately interpreted), I have a couple of concerns about the first part of the analysis: see major issues 1 and 2 below.

MAJOR ISSUES

1) One main finding of the study is that TAF and TDF have a similar effect on viral suppression at weeks 12 and 24, as described at the beginning of page 11. However, it looks like the difference between proportions has not been tested (or it has, but results have not been reported...). A two-sample test for proportions can be performed to show that, separately at weeks 12 and 24, proportions are not significantly different. Or, perhaps more elegantly, an ANOVA logistic model can be estimated to perform a simultaneous test on both the 12th and the 24th week.

Author’s Reply

We acknowledge that the differences in proportions (95% confidence intervals and p values) between the treatment groups were not computed, given the numerical similarity in responses. However, as requested, we have now performed a statistical comparison of treatment responses (TAF vs TDF) for both week 12 and week 24 endpoints (i.e. for % with HBV DNA <200,000, <20,000, and <29 IU/mL). Additional text describing the statistical methods employed has been added to the Methods on Page 9, and the results of the statistical analyses have been added to Results on Pages 12 and 13. There were no statistically significant differences seen in treatment responses for TAF vs TDF treatment at any of the 2 time points (i.e. weeks 12 and 24) or for any of the efficacy endpoints.

2) Figures 1 and 2 display the cumulative proportion of subjects who reach the suppression target at several weeks. I wonder why we should limit our attention to these two weeks when we could compare the two curves as a whole. For example, taking the complementary cumulative proportions (e.g. 1-p_w, where p_w is the cumulative proportion at week w), we obtain two survival curves that can be compared by a nonparametric logrank test, without the need of specific parametric assumptions. The authors should either perform this analysis or, alternatively, explain why weeks 12 and 24 are the only relevant weeks in this study.

Author’s Reply

For completeness and perspective, Figures 1 and 2 include comparative viral suppression rates over 48 weeks; however, as stated in the manuscript (and highlighted in the figures), the rates of suppression at weeks 12 and 24, which were derived from data generated from Studies GS-US-320-0110 (HBeAg-positive patients) and GS-US-320-0108 (HBeAg-negative patients), are considered most pertinent when applied to the setting of preventing mother-to-child transmission (MTCT) of HBV. Although current guidelines such as those from AASLD and APASL recommend 12 weeks of treatment starting at gestational weeks 28-32 for the prevention of MTCT in highly viremic mothers, the optimal duration of TDF (or TAF) treatment is largely unknown, and as pointed out in the manuscript, there are data from some studies to suggest 12 weeks of TDF treatment before delivery may not be sufficient for some patients (i.e. those with very high viral loads). In our analysis, by modeling the level of viremia reduction to <200,000 IU/mL at delivery, we aimed to investigate both a 12- and 24-week duration of TDF or TAF treatment to see whether a longer duration might represent a potential advantage for maternal viremia reduction. While the overall rates of viremia reduction to <200,000 IU/mL (Figure 1, Panel A), showed comparable results at weeks 12 and 24, we did demonstrate that for women of child bearing potential with baseline HBV DNA at or above 8 log10 IU/mL, higher proportions in both treatment groups achieved this end point at week 24 compared with week 12.

Reviewer #2: The manuscript entitled “Antiviral kinetics of TAF and TDF over 24 weeks in women of childbearing potential with chronic HBV” addresses an important topic in optimizing chronic HBV treatment among women of childbearing potential, as pregnant women were previously excluded from participation in clinical trials. The Introduction states that ‘several investigator-sponsored trials evaluating TAF for PMTCT of HBV are underway with results expected within a few years.” It is unclear to this reviewer why it takes several more years for these important trials to complete (given that the follow-up period for HBV is relatively short).

The present study provides reassuring data on early viral suppression in women of childbearing age and, as expected, TAF was well tolerated and showed less impact on renal and bone parameters than TDF. These findings are important and do support the ongoing clinical trials of TAF in pregnant women for HBV PMTCT. The reviewer notes that 4 coauthors are employed by the company (Gilead Sciences) and many coauthors have potential conflicts of interest; however, these potential conflicts have been transparently reported.

Following are several comments and suggestions for the authors:

(1) The authors mention current HBV care and treatment guidelines from the various professional liver societies around the world (with substantial variations noted in these regional guidelines). However, the authors fail to include the WHO HBV guidelines. On World Hepatitis Day (July 2020), WHO published updated guidelines for the PMTCT of HBV – which are highly relevant to the present article.

Author’s Reply 

The authors respectfully acknowledge that it was an oversight on our part not to cite or discuss the recently released and very pertinent WHO guidelines on antiviral prophylaxis for the prevention of mother-to-child transmission of HBV, and greatly appreciate the suggestion by the Reviewer to include this document in our manuscript. As our paper was submitted to PLOS One at a time coincident with the release of these guidelines last summer, this likely explains its omission. We have now included the WHO HBV PMTCT guidelines in the Introduction section as an additional citation (new Reference 18) on Page 5 in the sentence “Of these, TDF is preferred for prevention of MTCT (pMTCT) in all treatment guidelines given its high potency, proven efficacy, and acceptable safety profile in controlled trials [4, 9, 15-18].” We have also inserted content from the WHO Guidelines into the first paragraph of the Introduction on Page 6 as follows: “The World Health Organization (WHO) recommends that pregnant women testing positive for HBV infection (HBsAg-positive) with an HBV DNA ≥ 5.3 log10 IU/mL (≥ 200,000 IU/mL) receive TDF prophylaxis from the 28th week of pregnancy until at least birth for pMTCT [18].” Finally, reference to the WHO guidelines for pMTCT have also been added as text in the Discussion on Page 16 as follows: “ Waiting to initiate treatment between gestational weeks 28-32, as per the APASL, AASLD, and recently issued WHO guidelines [15, 17, 18], might be suboptimal for some women, particularly in the setting of very high HBV DNA levels and/or preterm delivery.”

(2) Two systematic reviews prepared for the latest 2020 WHO HBV guidelines have appeared in Lancet Infectious Diseases (published online August 2020; in hard copy, current issue): Boucheron et al; and Funk et al. These 2 systematic reviews provide relevant summary data for the Introduction and Discussion sections of your paper.

Author’s reply

The authors thank the Reviewer for this recommendation and have now added content from these 2 timely and pertinent systematic reviews to the manuscript. In the Introduction on Page 4, a sentence has been added to point out the recent publication by Boucheron et al regarding use of HBeAg testing to determine eligibility for antiviral prophylaxis (new Reference #8), and also in the introduction we have included text regarding the systematic review and meta-analysis by Funk et al (new reference 14) as well as in the inserted text in the first paragraph of the Discussion.

(3) First sentence “Worldwide, over 2 billion are infected with HBV …” (note this doesn’t seem accurately worded; over 2 billion people have become infected at some time during their lifetime). Chronic HBV infections worldwide are estimated by WHO at 257 million persons (see World Hepatitis Report, WHO, Geneva, 2017). The cited paper by Polaris – your reference 2 - is a relative outlier (‘up to 292 million’) among several published HBV modeling papers from 4-5 different groups. Your reference 1 (Schweitzer et al) estimated the number of people living with HBV infection at 240 million. The WHO estimate was a consensus estimate (involving many of the same investigator groups) – this reviewer recommends using the WHO (consensus) estimate.

Author’s Reply

The authors appreciate the clarification provided by the Reviewer and we have revised the wording in the first sentence of the Introduction accordingly. Also, it is appreciated that the citation of the Polaris data is considered by some to be an outlier. Therefore, at the suggestion of the Reviewer we have revised the estimated number to 257 million (the WHO figure) and replaced the Polaris citation with the WHO Global Hepatitis Report (Reference 2).

(4) Intro, page 6, last para: Could you mention anything on the price differential between TAF and TDF? In practice, this will be an important determinant of use.

Author’s Reply

While drug pricing is an important issue from a social and public health perspective, the authors respectfully decline the suggestion to include this in the manuscript as this is a secondary scientific analysis. Upon approval of TAF (Vemlidy) by the US FDA in 2016 for the treatment of chronic HBV its pricing has been on equal parity with that of TDF (Viread), so price has not been a key differentiator for their use. Finally, neither TDF nor TAF are currently indicated in their prescribing information for use in preventing mother-to-child transmission of HBV (rather the use of TDF is considered a public health measure driven by scientific data and treatment guidelines), therefore, any mention of drug pricing in the context of this paper we feel would not be appropriate.

(5) “Currently approved in over 70 countries worldwide, including China, Japan, S Korea, Taiwan, and Hong Kong.” This seems overly focused on one area of the world. How about Europe, Africa, S America? (or simply delete individual countries).

Author’s Reply

The Reviewer’s point is valid and as suggested we have now deleted any reference to specific countries wherein TAF is approved (Introduction, Page 6). 

(6) Page 7, line 3: ‘…same active moiety”. It seems accurately worded in this context. Just keep in mind, many people - like this reviewer - may not be familiar with ‘moiety’.

Author’s Reply

We have replaced the words “active moiety” with the term “active form” and included more detail as to what this refers to – specifically the active intracellular form of tenofovir which is tenofovir diphosphate.

(7) Page 7, line 5: “Investigator-sponsored trials …” - not sure I understand this. Do you mean “investigator-initiated”? Usually each trial has a financial sponsor (e.g., NIH, a company or a foundation).

Author’s Reply

As this descriptor is not necessarily needed in this context, we have deleted the term “investigator-sponsored” from the sentence. The authors intent in using this term was to point out that for these particular trials, Gilead Sciences, the manufacturer of TAF and TDF, is not the Sponsor of the studies, rather they are being conducted by various investigators who serve as the study sponsor while Gilead provides unrestricted grant funding for the research to be conducted.

(8) Page 8, line: “2 identically designed, prospective, randomized trials…” However, we learn that one trial focused on HBeAg-positive patients and the other on HBeAg-negative patients. Therefore, I think ‘identically designed’ is incorrect.

Author’s Reply

The only differences between the 2 trials in terms of study design were the study populations included (HBeAg-positive patients in Study GS-US-320-0110 and HBeAg-negative patients in Study GS-US-320-0108), and by extension, the number of patients enrolled in each study due to the estimated sample sizes required; however, we can appreciate that technically in this sense, the designs were not truly identical. Thus, the term “identically designed...” has been deleted from the sentence. Instead, we have added new text to the end of the following sentence stating the following point “otherwise the 2 studies were identical in design.” 

(9) The study design included women of childbearing potential as ‘females 18-49 years of age’ – with a median age of 35 years. Thus, the average age in your study is higher than the average age of pregnant women in most countries (around 25-26 years of age) though this is changing. Not sure if this 10-year age difference might change any of the findings – I suggest including it among the study limitations.

Author’s Reply

We agree the age difference could be considered an issue as suggested, although we do not have any data to suggest that viral load responses for TAF and TDF are influenced by age, particularly in younger, pre-menopausal women. However, it is acknowledged that this should be mentioned as a potential study limitation. Thus, we have added a sentence to the study limitations paragraph in the Discussion to make this point.

(10) Discussion: page 13, line 12-13: Does your study provide any data on HBV viral load suppression ‘less than 12 weeks’ after initiation of HBV treatment (either TAF or TDF)? Such data would be very helpful.

Author’s Reply

As shown in Figure 1, HBV DNA suppression was also assessed earlier than Week 12 – at Weeks 4 and 8. As expected, the suppression rates are lower following initiation of treatment than those seen at Weeks 12 and 24, and the figure does provide the reader with a frame of reference for these earlier suppression rates which we believe is sufficient.

(11) Discussion, end of para 1: Again, it would be helpful to include the updated 2020 WHO HBV guidelines in your Discussion. In many countries (e.g., in Africa, Latin America, many countries in Asia outside China), these would be the ones used.

Overall, this is an important paper and the findings among women of childbearing potential appear quite unique in the HBV literature.

Author’s Reply

As discussed earlier in our response to the Reviewer we have now included reference to the updated WHO HBV guidelines in the Discussion.

Here are a few minor issues:

1) Reference 3 lacks information on year and pages.

Author’s Reply

Thank you for pointing this out. The journal citation has been corrected based on information found in PubMed to: PLoS One 2017 Jun 2;12(6):e0178671. doi: 10.1371/journal.pone.0178671.

2) Table headings need more information: for example, they should include information about the study population/year of the study cohorts included. Currently, the table headings cannot be interpreted or ‘stand on their own’.

Author’s Reply

We have now added information to all table headings to indicate data sources.

---

## [Editor Report · Decision Letter 1]

29 Apr 2021

Antiviral kinetics of tenofovir alafenamide and tenofovir disoproxil fumarate over 24 weeks in women of childbearing potential with chronic HBV

PONE-D-20-22518R1

Dear Dr. Pan,

We’re pleased to inform you that your manuscript has been judged scientifically suitable for publication and will be formally accepted for publication once it meets all outstanding technical requirements.

Kind regards,

Marc Bulterys

Guest Editor

PLOS ONE
---

## [Editor Report · Acceptance letter]

4 May 2021

PONE-D-20-22518R1 

Antiviral kinetics of tenofovir alafenamide and tenofovir disoproxil fumarate over 24 weeks in women of childbearing potential with chronic HBV 

Dear Dr. Pan:

I'm pleased to inform you that your manuscript has been deemed suitable for publication in PLOS ONE. Congratulations! Your manuscript is now with our production department. 

Kind regards, 

on behalf of

Dr. Marc Bulterys 

Guest Editor

PLOS ONE